# Taste 2 Receptor Is Involved in Differentiation of 3T3-L1 Preadipocytes

**DOI:** 10.3390/ijms23158120

**Published:** 2022-07-23

**Authors:** Shunsuke Kimura, Ai Tsuruma, Eisuke Kato

**Affiliations:** 1Graduate School of Agriculture, Hokkaido University, Kita-ku, Sapporo 060-8589, Japan; gpzshun2239r@gmail.com (S.K.); story-seller-1019@eis.hokudai.ac.jp (A.T.); 2Japan Society for the Promotion of Science (JSPS), Chiyoda-ku, Tokyo 102-0083, Japan; 3Division of Fundamental AgriScience and Research, Research Faculty of Agriculture, Hokkaido University, Kita-ku, Sapporo 060-8589, Japan

**Keywords:** taste 2 receptor, bitter taste receptor, epicatechin, 3T3-L1 adipocyte, differentiation, lipid accumulation

## Abstract

Expression of taste 2 receptor (T2R) genes, also known as bitter taste receptor genes, has been reported in a variety of tissues. The white adipose tissue of mice has been shown to express *Tas2r108*, *Tas2r126*, *Tas2r135*, *Tas2r137*, and *Tas2r143*, but the function of T2Rs in adipocytes remains unclear. Here, we show that fasting and stimulation by bitter compounds both increased *Tas2r* expression in mouse white adipose tissue, and serum starvation and stimulation by bitter compounds both increased the expression of *Tas2r* genes in 3T3-L1 adipocytes, suggesting that T2Rs have functional roles in adipocytes. RNA sequencing analysis of 3T3-L1 adipocytes stimulated by epicatechin, the ligand of Tas2r126, suggested that this receptor may play a role in the differentiation of adipocytes. Overexpression of *Tas2r126* in 3T3-L1 preadipocytes decreases fat accumulation after induction of differentiation and reduces the expression of adipogenic genes. Together, these results indicate that Tas2r126 may be involved in adipocyte differentiation.

## 1. Introduction

Taste 2 receptors (T2Rs) sense bitter taste in oral cavities and are members of the G-protein coupled receptor family. Interestingly, T2R genes are widely expressed in extra-oral tissues, including the airway, gastrointestinal tract, brain, heart, liver, testis, and adipose tissues [1], and T2Rs have specific functions in different tissues. For example, the T2Rs in airway tissue can sense chemicals secreted from Gram-negative bacteria, leading to activation of nitrogen oxide synthase and the production of nitrogen oxide, which then induces a direct bactericidal effect and activation of villus movement to eliminate bacteria from airways [2]. T2Rs in stomach tissue can detect caffeine and stimulate the secretion of gastric acid [3]. T2Rs in intestinal tuft cells, a rare type of intestinal epithelial cell, can detect secretory products from the parasitic helminth *Trichinella spiralis* and stimulate the release of interleukin 25 to activate the immune response [4].

In a previous study, we showed that *Tas2r108*, *Tas2r126*, *Tas2r135*, *Tas2r137*, and *Tas2r143* are expressed in mouse adipose tissue [5]; however, the functions of these receptors in adipocytes are still unclear. Several reports have suggested that T2Rs may have some role in adipocytes. For example, Chupeerach et al. found that single nucleotide polymorphisms in T2R genes were related to the risk of obesity development in humans [6]. Furthermore, the bitter-tasting compounds epigallocatechin gallate [7], resveratrol [8], nomilin [9], and isohumulone [10] have been reported to have anti-obesity effects in diet-induced obese mice. Together, these findings indicate that T2Rs may regulate lipid metabolism in adipocytes.

In this study, we aimed to explore the function of T2Rs in adipocytes. We investigated changes in T2R gene expression triggered by several environmental factors in epididymal and inguinal white adipose tissues (eWAT and iWAT) of C57BL/6J mice as well as in 3T3-L1 adipocytes. Comprehensive analysis of gene expression changes in 3T3-L1 adipocytes stimulated by bitter compounds and overexpression of T2R genes in 3T3-L1 preadipocytes provided insights into the function of T2Rs in adipocytes.

## 2. Results

### 2.1. Tas2r Expression in 3T3-L1 Adipocytes Responds to External Stimuli

Expression of *Tas2r108*, *Tas2r126*, *Tas2r135*, *Tas2r137*, and *Tas2r143* in mouse white adipose tissue (WAT) and mouse-derived 3T3-L1 adipocytes has been reported previously [5]. To determine if the encoded T2Rs are functional in adipocytes, the responses of these *Tas2r* genes to external stimuli were evaluated in 3T3-L1 adipocytes under the following three conditions: (1) induction of differentiation, (2) deprivation of medium components, and (3) stimulation by bitter compounds.

We found that induction of differentiation increased the expression of all five *Tas2r* genes (*Tas2r108*, *Tas2r126*, *Tas2r135*, *Tas2r137*, *Tas2r143*) in 3T3-L1 adipocytes (Figure 1). Deprivation of amino acids had no effect on the expression of the *Tas2r* genes, whereas deprivation of serum increased the expression of *Tas2r108*, *Tas2r126*, and *Tas2r137* (Figure 2). Among the bitter compounds that are known ligands of mouse T2Rs (Table 1), stimulation by epicatechin, camphor, and allyl isothiocyanate upregulated the expression of *Tas2r126*, *Tas2r135*, and *Tas2r143* and slightly increased the expression of *Tas2r108* and *Tas2r137* (Figure 3). We chose concentrations of the bitter compounds that were above the reported thresholds [11] and confirmed they did not reduce cell viability (Appendix A). These results show that the *Tas2r* genes responds to environmental stimuli, which suggests that T2Rs may have functions in 3T3-L1 adipocytes.

### 2.2. Tas2r Expression in Mouse WAT Responds to External Stimuli

To evaluate changes in the expression of *Tas2r* genes in mouse WAT, we stimulated mouse eWAT and iWAT with epicatechin, the ligand of Tas2r126. We found that, although the response in mouse WAT was different from that in 3T3-L1 adipocytes, epicatechin upregulated the expression of *Tas2r108* and *Tas2r137* in both eWAT and iWAT (Figure 4). When mice were fasted for 16 h, the expression of *Tas2r108* increased and the expression of *Tas2r137* slightly increased in iWAT but not in eWAT (Figure 5). These results indicate that T2Rs may have functions in mouse WAT.

### 2.3. RNA Sequencing (RNA-Seq) Analysis of 3T3-L1 Adipocytes Stimulated by Bitter Compounds

To evaluate the function of T2Rs in adipocytes, we performed RNA-seq analysis of 3T3-L1 adipocytes stimulated by bitter compounds. Mature 3T3-L1 adipocytes were stimulated by quinine (ligand of Tas2r108, 126, 137) or epicatechin (ligand of Tas2r126) for 1 h, and gene expression was compared between stimulated and non-stimulated adipocytes. Genes that were differentially expressed were selected for gene ontology (GO) functional enrichment analysis [12,13]. In the quinine-stimulated adipocytes, the clustering analysis showed that the expression levels of the *Tas2r* genes were similar to those in the non-stimulated adipocytes, and therefore no further analysis was performed for these cells (Appendix A). The upregulated genes in the epicatechin-stimulated adipocytes were annotated with the “fat cell differentiation” term under the biological process GO category (Figure 6A,B). These genes were further analyzed to find out if T2Rs are involved in lipid metabolism in adipocytes. Only a few of the genes in the epicatechin-stimulated adipocytes were downregulated and only six GO terms were assigned to them. These genes were not selected for further analysis (Appendix A).

The upregulated genes annotated as related to “fat cell differentiation” included the transcription factors early growth response 2 (*Egr2*) and nuclear receptors *Nr4a1*, *Nr4a2*, and *Nr4a3*. These genes were re-analyzed by real-time quantitative PCR (RT-qPCR) to confirm their upregulated expression in 3T3-L1 adipocytes by epicatechin stimulation (Figure 6C).

### 2.4. Overexpression of Tas2r108 or Tas2r126 Reduced Differentiation of 3T3-L1 Preadipocytes

The *Egr2*, *Nr4a1*, *Nr4a2*, and *Nr4a3* genes that were upregulated in epicatechin-stimulated 3T3-L1 adipocytes have been reported previously to influence adipocyte differentiation [14,15,16,17]. To determine if the upregulation of these genes in 3T3-L1 adipocytes is mediated by T2Rs, we overexpressed *Tas2r126* in 3T3-L1 preadipocytes and evaluated the effect on differentiation. We also overexpressed *Tas2r108* in 3T3-L1 preadipocytes to check whether it had the same function as *Tas2r126*.

*Tas2r108*- and *Tas2r126*-overexpressing preadipocytes were constructed by transfecting a plasmid vector carrying the *Tas2r108* (NM_020502.1) or *Tas2r126* (NM_207028.1) mRNA sequence. After transfection, the cells were treated with hygromycin B to select the overexpressing preadipocytes, and the selected cells were used for the experiment. Overexpression of the *Tas2r* genes in the preadipocytes was confirmed by RT-qPCR (*Tas2r108* and *Tas2r126* expression were 79 and 40 times, respectively, compared with that in the control).

Differentiation of the *Tas2r108*- and *Tas2r126*-overexpressing preadipocytes was induced with or without epicatechin. Lipid droplets were stained using AdipoRed™ assay reagent (Lonza K.K., Tokyo, Japan). We found that overexpression of *Tas2r126* decreased the accumulation of lipids compared with their accumulation in the control, and the presence of epicatechin during differentiation further reduced the accumulation of lipids (Figure 7). Overexpression of *Tas2r108* also gave similar results. RT-qPCR analysis of *Pparg* and *Cebpa* expression, the two adipogenic genes responsible for differentiation of adipocytes, showed that their expression decreased in the *Tas2r108*- and *Tas2r126*-overexpressing cells before and after induction of differentiation, suggesting decreased accumulation of lipids may be due to reduced differentiation of the cells (Figure 8).

## 3. Discussion

Human and mouse WAT are known to express T2R genes, but whether these genes are functional and their functions have not yet been elucidated [5,18].

We found that the expression of *Tas2r* genes increased in mouse WAT that was stimulated by bitter compounds or after fasting (Figure 4 and Figure 5). Increased expression of *Tas2r* genes was also detected after induced differentiation of 3T3-L1 adipocytes, stimulation by bitter compounds, and serum starvation (Figure 1, Figure 2 and Figure 3). The responses of *Tas2r* genes to these stimuli imply that T2Rs are functional in WAT and 3T3-L1 adipocytes. T2Rs in the tuft cells of the intestine respond to parasite secretory products, and increased expression of *Tas2r* genes was detected upon stimulation [4]. Fasting/refeeding has also been reported to modulate the expression of *Tas2r138* in mouse stomachs [19], and T2Rs in stomach tissue have been reported to regulate gastric acid secretion in humans [3]. The response of *Tas2r* genes to fasting also supports the presumption that T2Rs are functional in WAT. However, because WAT consists not only of adipocytes but many other types of cells, these other cells may influence the results.

Interestingly, the changes in the expression of the *Tas2r* genes in response to fasting were different in eWAT and iWAT (Figure 5). We have no clear explanation for this difference. One speculation is that the tissues surrounding iWAT may secret a compound that stimulates *Tas2r* gene expression upon fasting, causing a difference between eWAT and iWAT.

RNA-seq analysis of 3T3-L1 adipocytes suggested that the mouse T2R Tas2r126 is related to fat cell differentiation (Figure 6B). The 3T3-L1 preadipocytes overexpressing *Tas2r108* and *Tas2r126* showed reduced differentiation, indicating that the encoded T2Rs are involved in adipocyte differentiation. The addition of epicatechin to the overexpressing 3T3-L1 preadipocytes further reduced the accumulation of lipids (Figure 7); however, the non-differentiated wild-type 3T3-L1 adipocytes also showed reduced lipid accumulation with epicatechin stimulation, and therefore, it could not be confirmed whether epicatechin is involved in adipocyte differentiation via T2R. Furthermore, although both *Tas2r108* and *Tas2r126* overexpression suppressed the differentiation of 3T3-L1 preadipocytes, their effects on the adipogenic genes *Pparg* and *Cebpa* were not completely the same. Overexpression of *Tas2r108* strongly suppressed the expression of *Cebpa,* but overexpression of *Tas2r126* had a weaker effect. However, this may be because overexpression of *Tas2r108* was higher than that of *Tas2r126* (*Tas2r108* 79 times and *Tas2r126* as follows: 40 times their expression in wild type).

Avau et al. [20] reported that the treatment of 3T3-F442 preadipocytes with the bitter compounds quinine and denatonium benzoate inhibited their differentiation and suggested, but did not show, that T2Rs may be involved in adipocyte differentiation. This is consistent with our finding that epicatechin inhibited the differentiation of 3T3-L1 preadipocytes (Figure 7). Furthermore, our data show that the expression of *Tas2r108* and *Tas2r126*, targets of quinine and epicatechin, influenced differentiation and may be involved in the ability of bitter compounds to inhibit the differentiation of adipocytes.

Ning et al. [21] reported that knockdown of *Tas2r106* reduced lipid accumulation and decreased the expression of adipogenic genes in 3T3-L1 adipocytes. Although overexpression and knockdown methods have opposite effects, and the *Tas2r* gene was different, their findings do not contradict our results. Thus, we concluded that T2Rs are involved in adipocyte differentiation.

Our study has several limitations. The major limitation is that the presence of T2R proteins in WAT and 3T3-L1 adipocytes has not yet been confirmed, mainly because of the lack of a specific antibody against mouse T2R. We are currently working on this. Another limitation is that overexpression of *Tas2r108* and *Tas2r126* was performed in 3T3-L1 preadipocytes, whereas *Tas2r* gene expression was higher in mature adipocytes than it was in preadipocytes (Figure 1). Ideally, T2Rs should be studied in mature adipocytes rather than in preadipocytes. Thus, a study using mature adipocytes is needed to further explore the functions of T2Rs in adipocytes. Lastly, this study was conducted mainly in 3T3-L1 adipocytes, which are model cells of adipocytes, but they are different from the adipocytes in mouse WAT. Because the response to stimulation by epicatechin was different in 3T3-L1 adipocytes and mouse WAT (Figure 2 and Figure 4), the function of T2Rs in 3T3-L1 adipocytes and mouse WAT may be different as well. Therefore, studies using mouse adipose tissue are essential to confirm the function of T2Rs in adipocytes.

To conclude, our results suggest that *Tas2r* genes are expressed in mouse WAT and 3T3-L1 adipocytes, and the function of the mouse T2Rs is related to the regulation of adipocyte differentiation. Additional studies are required to further understand the function of T2Rs in our bodies.

## 4. Materials and Methods

### 4.1. Cell Culture

The 3T3-L1 preadipocytes (JCRB9014) were obtained from the Japanese Collection of Research Bioresources Cell Bank in Osaka, Japan. The cells were cultured in growth medium [Dulbecco’s modified Eagle medium (DMEM) supplemented with 10% fetal bovine serum (FBS) and antibiotics (100 units/mL penicillin, 100 µg/mL streptomycin, and 50 µg/mL gentamicin)] at 37 °C in a 10% carbon dioxide atmosphere.

Mature 3T3-L1 adipocytes were prepared as follows. After the cells reached confluence (Day 0), the medium was replaced with growth medium supplemented with 0.5 mM 3-isobutyl-1-methylxanthine, 0.25 µM dexamethasone, and 10 µg/mL insulin and the cells were cultured for 2 days. On Day 2, the medium was replaced with growth medium supplemented with 5 µg/mL insulin and the cells were cultured for 4 more days. On Day 6, the medium was replaced with growth medium, and the cells were cultured for a further 4 days. On Day 10, differentiation of the cells was confirmed by microscopic observation. These cells were used for the experiments.

For bitter compound stimulation, each compound was dissolved in dimethyl sulfoxide (DMSO) or water and diluted in growth medium without antibiotics. The medium of the mature 3T3-L1 adipocytes was replaced with medium containing one of the bitter compounds and the cells were incubated for 1 h. The DMSO concentration was kept below 0.1% and same concentration of DMSO was added to medium of the control cells.

For serum and amino acid deprivation, the medium of the mature 3T3-L1 adipocytes was replaced with DMEM, DMEM without amino acids supplemented with 10% FBS, or DMEM without amino acids, and the cells were incubated for 1 h.

### 4.2. Animal Experiments

C57BL/6J male mice (6 weeks old) were housed in an air-conditioned room at 23  ±  2 °C with a light period from 8:00 a.m. to 8:00 p.m. The mice were fed a high-fat diet (Research Diet Inc., New Brunswick, NJ, USA, D12492) for 8–10 weeks, then fasted for 16 h. The mice were euthanized by cervical dislocation following sevoflurane anesthesia, and eWAT and iWAT were collected. Fasting was skipped to compare fasted and non-fasted WAT.

For bitter compound stimulation, the collected WAT samples were washed with phosphate-buffered saline and incubated in DMEM supplemented with 10% FBS containing 2.5 mM epicatechin for 3 h.

### 4.3. RT-qPCR

Total RNA was extracted from 3T3-L1 adipocytes using a FastGene™ RNA Premium kit (Nippon Gene Co., Ltd., Tokyo, Japan). Total RNA was extracted from the WAT using RNeasy Lipid Tissue Mini Kit (Qiagen, Tokyo, Japan). Reverse transcription was performed using ReverTra Ace qPCR RT Master Mix with gDNA Remover (Toyobo Co., Ltd., Osaka, Japan) and a T100 thermal cycler (Bio-Rad Laboratories, Inc., Hercules, CA, USA). The RT-qPCRs were performed using a Thermal cycler dice real-time system (Takara Bio Inc., Shiga, Japan) and GeneAce SYBR qPCR Mix α (Nippon Gene Co., Ltd.) with the primers listed in Table 2. The qPCR reaction conditions were as follows: enzyme activation at 95 °C for 10 min, followed by 45 cycles at 95 °C for 30 s and 60 °C for 60 s.

### 4.4. RNA-Seq Analysis

RNA-seq was performed and analyzed by Rhelixa Co. (Tokyo, Japan) using an Illumina NovaSeq 6000 (Illumina, Inc., San Diego, CA, USA). Total RNA was extracted from the 3T3-L1 adipocytes using RNeasy Plus mini kit (Qiagen, Tokyo, Japan), analyzed using a Bioanalyzer and reads with RNA integrity numbers >9.0 were used for the analysis. Differentially expressed genes were extracted (*p* < 0.05) and GO analysis was performed.

### 4.5. Overexpression of Tas2r Genes

The *Tas2r108* (NM_020502.1) and *Tas2r126* (NM_207028.1) sequences with the bases that encode 45 amino acids of rat somatostatin receptor 3 [22] and HSV glycoprotein epitope tag attached at their 5′ and 3′ ends, respectively, were cloned into a pcDNA™3.1/Hygro(+) vector (Thermo Fisher Scientific Inc., Waltham, MA, USA). The vector was transfected into 3T3-L1 preadipocytes using Lipofectamine 3000 reagent (Thermo Fisher Scientific Inc.) and Opti-MEM™ I reduced serum medium (Thermo Fisher Scientific Inc.). Three days after transfection, hygromycin B (0.5 mg/mL) was added to the medium and the cells were selected for 1 week. The selected cells were used for the experiments.

### 4.6. Lipid Accumulation Assay

3T3-L1 preadipocytes were seeded in a 48-well plate and cultured. After reaching confluence, induced differentiation of the cells was performed as described in Section 4.1. On Day 6, the cells are stained using AdipoRed™ assay reagent (Lonza K.K., Sagamihara, Japan) and fluorescence was measured using a Synergy™ MX microplate reader (Agilent Technologies, Inc., Santa Clara, CA, USA) to detect lipid accumulation.

### 4.7. Statistics

All the experiments were repeated at least twice. Representative data are expressed as mean ± standard error of the mean (SEM) in the figures. Data were analyzed using GraphPad Prism software (v9.3.1) with the statistical methods indicated in the figure legends.

## Figures and Tables

**Figure 1 ijms-23-08120-f001:**
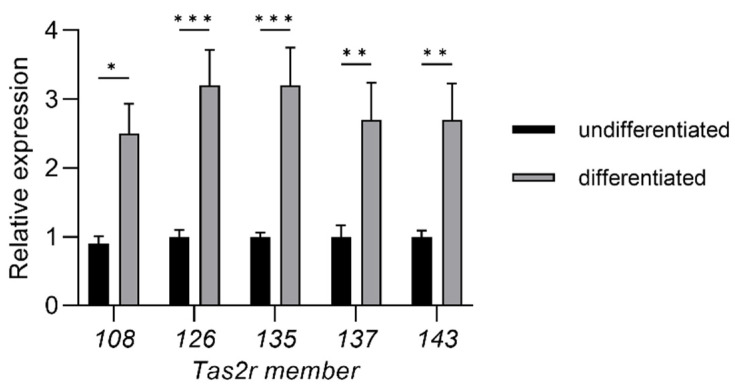
Induction of differentiation increased the expression of *Tas2r* genes in 3T3-L1 adipocytes. Total RNA was extracted from 3T3-L1 preadipocytes (undifferentiated) or mature adipocytes (differentiated). *Tas2r* expression was determined by real-time quantitative PCR (RT-qPCR) analysis. The actin beta gene (*Actb*) was used as the reference gene. N = 6, * *p* < 0.05, ** *p* < 0.01, *** *p* < 0.001 (differentiated vs. undifferentiated, Sidak test).

**Figure 2 ijms-23-08120-f002:**
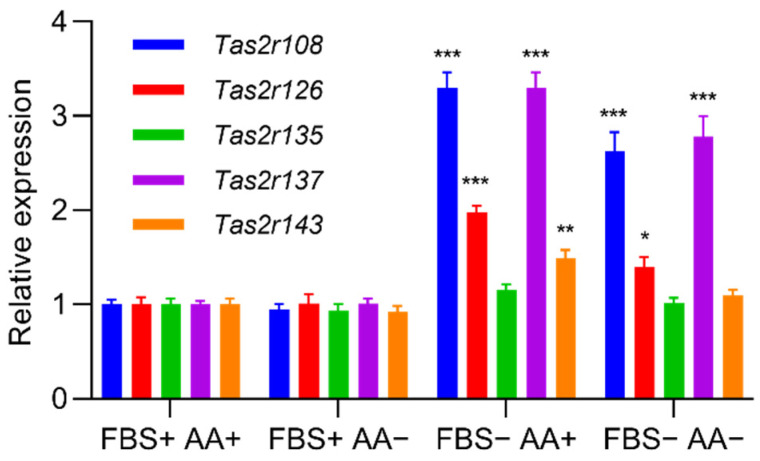
Deprivation of serum increased the expression of *Tas2r* genes in 3T3-L1 adipocytes. The 3T3-L1 adipocytes were cultured in the medium with (+) or without (−) fetal bovine serum (FBS) and amino acids (AA) for 1 h. *Tas2r* expression was determined by RT-qPCR analysis. *Actb* was used as the reference gene. N = 6, * *p* < 0.05, ** *p* < 0.01, *** *p* < 0.001 (vs. FBS+AA+, Dunnett test).

**Figure 3 ijms-23-08120-f003:**
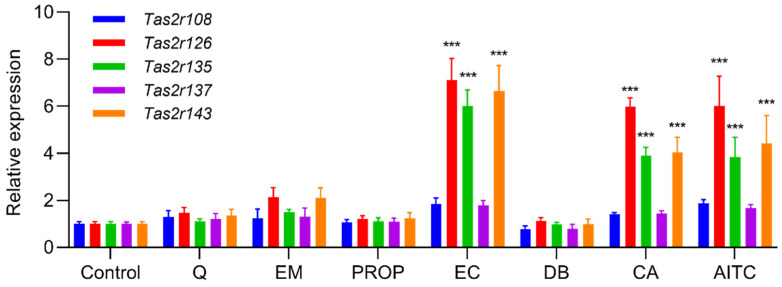
Stimulation by bitter compounds increased the expression of *Tas2r* genes in 3T3-L1 adipocytes. Mature 3T3-L1 adipocytes were stimulated by quinine (Q, 50 µM), emetine (EM, 150 µM), 6-propyl-2-thiouracil (PROP, 5 mM), epicatechin (EC, 2.5 mM), denatonium benzoate (DB, 500 µM), camphor (CA, 5 mM), or allyl isothiocyanate (AITC 150 µM) for 1 h. *Tas2r* expression was determined by RT-qPCR analysis. *Actb* was used as the reference gene. N = 6, *** *p* < 0.001 (compound vs. control, Dunnett test).

**Figure 4 ijms-23-08120-f004:**
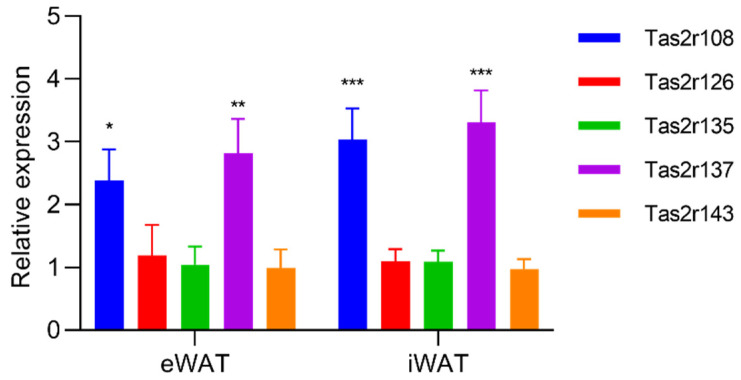
Stimulation by epicatechin upregulated the expression of *Tas2r* genes in mouse white adipose tissue (WAT). Mouse epididymal WAT (eWAT) and inguinal WAT (iWAT) were stimulated with epicatechin (EC, 2.5 mM) for 3 h. *Tas2r* expression was determined by RT-qPCR analysis. *Actb* was used as the reference gene. N = 6, * *p* < 0.05, ** *p* < 0.01, *** *p* < 0.001 (stimulated vs. control, *t*-test).

**Figure 5 ijms-23-08120-f005:**
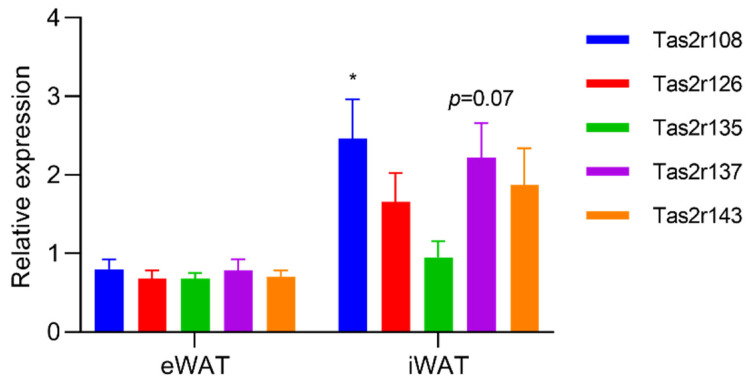
Fasting upregulated the expression of *Tas2r* genes in mouse iWAT. Mice were fasted for 16 h, then eWAT and iWAT were collected. *Tas2r* expression was determined by RT-qPCR analysis. *Actb* was used as the reference gene. N = 6, * *p* < 0.05 (fasted vs. unfasted, Sidak test).

**Figure 6 ijms-23-08120-f006:**
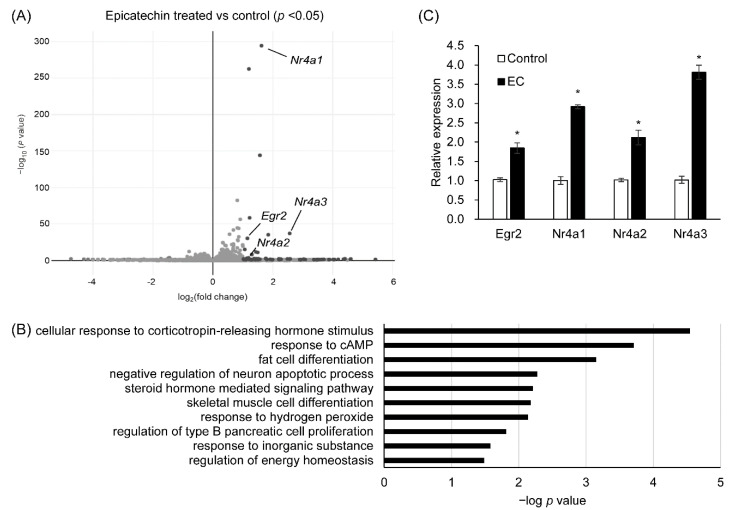
Expression and gene ontology (GO) analysis of genes that were upregulated in epicatechin-stimulated 3T3-L1 adipocytes. The 3T3-L1 adipocytes were stimulated with epicatechin (2.5 mM) for 1 h, total RNA was extracted, and RNA-seq analysis was performed. Volcano plot (**A**) and GO analysis (**B**) of the upregulated genes. Confirmation of the expression of the selected genes in the epicatechin-stimulated adipocytes by RT-qPCR (**C**). N = 3 for RNA-seq analysis, N = 6 for RT-qPCR, * *p* < 0.05 (stimulated vs. unstimulated, *t*-test).

**Figure 7 ijms-23-08120-f007:**
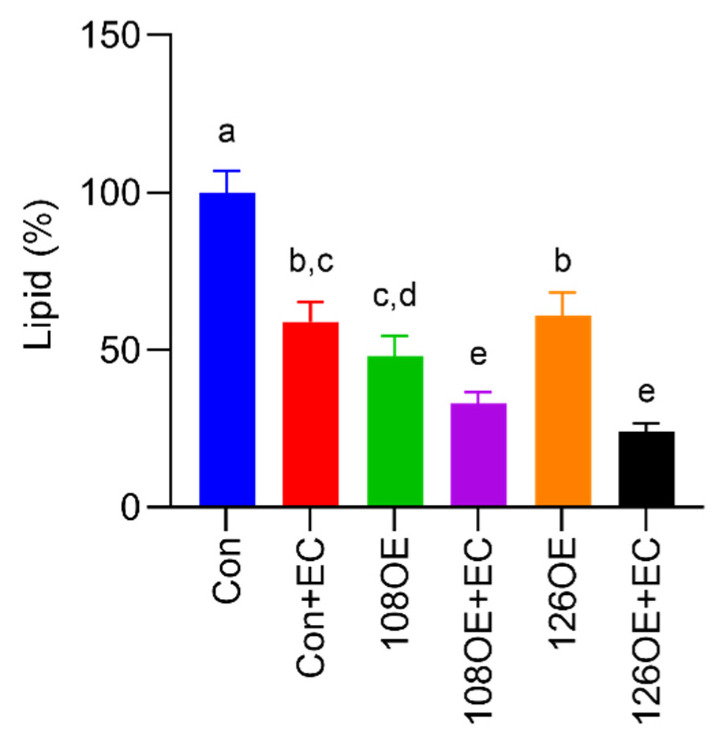
Accumulation of lipids after differentiation of *Tas2r108*- or *Tas2r126*-overexpressing 3T3-L1 preadipocytes. Differentiation of 3T3-L1 preadipocytes overexpressing *Tas2r108* (108OE) or *Tas2r126* (126OE) was induced with or without epicatechin (EC, 1.0 mM). The cells were stained using AdipoRed™ assay reagent. Con: control, N = 6, different letters (a,b,c,d,e) indicate significant differences compared with the other bars (*p* < 0.05, Tukey test).

**Figure 8 ijms-23-08120-f008:**
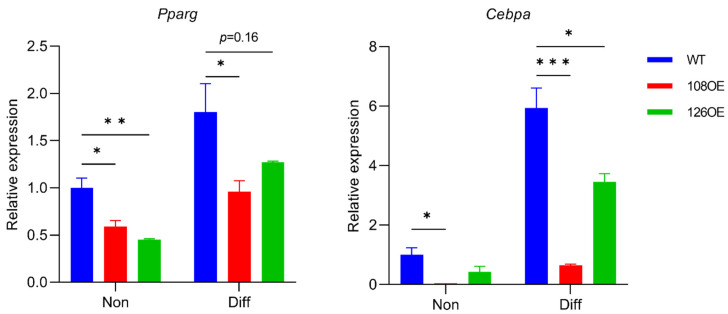
Expression of adipogenic genes *Pparg* and *Cebpa* was reduced in 3T3-L1 adipocytes overexpressing *Tas2r108* (108OE) or *Tas2r126* (126OE). *Pparg* and *Cebpa* expression before (Non) and after (Diff) induction of differentiation compared with their expression in non-differentiated wild-type (WT) cells. N = 3, * *p* < 0.05, ** *p* < 0.01, *** *p* < 0.001, Dunnett test.

**Table 1 ijms-23-08120-t001:** Bitter compounds and threshold concentrations for activation of mouse taste 2 receptors (T2Rs) [11].

Compound (Abbreviation)	Target T2R	Threshold (μM)
quinine (Q)	Tas2r108, Tas2r126, Tas2r137	10
emetine (EM)	Tas2r108	30
6-propyl-2-thiouracil (PROP)	Tas2r108, Tas2r135, Tas2r137	1000
epicatechin (EC)	Tas2r126	1000
denatonium benzoate (DB)	Tas2r135	100
allyl isothiocyanate (AITC)	Tas2r135	300
camphor (CA)	Tas2r137	1000

**Table 2 ijms-23-08120-t002:** Primers used in this study.

Target	Forward	Reverse
*Actb*	TACGACCAGAGGCATACAG	GCCAACCGTGAAAAGATGAC
*Tas2r108*	AACAGGACCAGCTTTTGGAATC	GAGGAAACAGATCATCAGCCTCAT
*Tas2r126*	GCTCAGCGTCCTGTTCTGTA	CAACGCTGGGAATCTCCACT
*Tas2r135*	GAACTTCGGGATGTCTGGGC	TATGGTGTGTTGCTGGCAGA
*Tas2r137*	TTCTACTGCCTGAAAATAGCCAGTT	AACAACCACTCTAGAAGCTCTCCATT
*Tas2r143*	TCCCAGTTAGTTCCCAGGCT	AAGTTCCCGGTGGCTGAAAT
*Pparg*	GTACTGTCGGTTTCAGAAGTGCC	ATCTCCGCCAACAGCTTCTCCT
*Cebpa*	GCAAAGCCAAGAAGTCGGTGGA	CCTTCTGTTGCGTCTCCACGTT
*Egr2*	CCTTTGACCAGATGAACGGAGTG	CTGGTTTCTAGGTGCAGAGATGG
*Nr4a1*	GTGCAGTCTGTGGTGACAATGC	CAGGCAGATGTACTTGGCGCTT
*Nr4a2*	CCGCCGAAATCGTTGTCAGTAC	TTCGGCTTCGAGGGTAAACGAC
*Nr4a3*	ACGCCGAAACCGATGTCAGTAC	CTCCTGTTGTAGTGGGCTCTTTG

## Data Availability

The RNA-seq analysis data are available in the Genomic Expression Archive (https://www.ddbj.nig.ac.jp/gea/index.html, accessed on 21 July 2022). Accession number: DRA014547.

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
