# Peer review of "Taste 2 Receptor Is Involved in Differentiation of 3T3-L1 Preadipocytes"

_ijms, 2022, doi:10.3390/ijms23158120_

Round 1
Reviewer 1 Report
In this manuscript, Kimura et al. address the expression patterns of T2Rs in adipocytes following fasting or stimulation by bitter molecules. The study is interesting and well designed, the results are novel and of interest. I have several minor comments though:
- based on what criterion were the concentrations of bitter substances selected for the stimulation?
- how exactly was the tissue collectede from the mice? Were the animals sacrified prior to the collection?
- technical equipment is missing in several cases. For instance, what machine was used for the fluorescence detection in the lipid accumulation assay?
- the expression of T2Rs was analyzed on a genetic level. have the authors considered to validate changes in the expression patterns with a proteomic tool, for instance Western blots? This could be discussed.
- Minor grammar errors and typos should be corrected.
Reviewer 2 Report
In this study, the authors investigated the effect of bitter compounds and fasting on Tas2r gene expression in white adipose tissue and 3T3-L1 adipocytes. They found that that both these factors increased the expression of Tas2r in mouse adipose tissue and 3T3-L1 adipocytes. Moreover, they showed that overexpression of Tas2r in differentiated 3T3-L1 cells decreased fat accumulation and adipogenic genes expression. The authors conclude that Tas2r are functional in adipose tissue and may have a role in differentiation of adipocytes.
Before accepting these conclusions, a some facts should be clarified
1) There are few published studies on T2R in adipocytes. Therefore, authors should cite the published study: Avau, B., Bauters, et al. (2015). The Gustatory Signaling Pathway and Bitter Taste Receptors Affect the Development of Obesity and Adipocyte Metabolism in Mice. PloS one, 10(12), e0145538. https://doi.org/10.1371/journal.pone.0145538 PMID: 26692363. In addition, authors should explain how their results are consistent with this study and what is new about their study. Please note that the referee is not a co-author of this study.
2)Figure 4 and 5: Epicatechin stimulation upregulated Tas2r gene expression in white adipose tissue from epididymal and inguinal fat pads. In contrast, starvation upregulated Tas2r only in inguinal adipose tissue. How can this different response be explained even though the metabolic functions of these tissues are identical?
3)Line 152: Figure 7: The legend is unnecessary in the chart when the groups are listed below individual columns. The indication of statistical significance is not clear. It is not stated what the individual letters mean. It is also unclear why *p<0.05 is listed in the text below the image when it is not listed in the image.
4)Methods Section, Lines 238-240: The method used to monitor the stimulation of Tas2r genes expression by epicatechin in mouse adipose tissue is not clearly described. What does it mean that adipose tissue has been "soaked" with epicatechin? Does that mean that the tissue it was incubated for 3 hours in a medium that contained epicatechin?
5)For further studies, the incubation of adipose tissue in a suitable medium (Krebs-Ringer bicarbonate) could be used to monitor the utilization of substrates for lipogenesis (14C -Glucose) or release fatty acid from adipose tissue relative to Tas2r gene expression. This would provide direct evidence that T2Rs are functionally expressed in adipocytes.
that T2Rs are functionally expressed in adipocytes
6)The authors should discuss whether the high concentrations of bitter compounds used to stimulate Tas2r genes in vitro can be hardly reached in human tissues.
Author Response
Please see the attachment。
